# Acute Pyelonephritis with Bacteremia in an 89-Year-Old Woman Caused by Two Slow-Growing Bacteria: *Aerococcus urinae* and *Actinotignum schaalii*

**DOI:** 10.3390/microorganisms11122908

**Published:** 2023-12-02

**Authors:** Laurène Lotte, Claire Durand, Alicia Chevalier, Alice Gaudart, Yousra Cheddadi, Raymond Ruimy, Romain Lotte

**Affiliations:** 1Department of Biology, Cannes General Hospital, 06400 Cannes, France; la.lotte@ch-cannes.fr; 2Department of Infectious Diseases, Nice University Hospital, 06003 Nice, France; durand.c@chu-nice.fr; 3Department of Bacteriology, Nice University Hospital, 06003 Nice, France; chevalier.a3@chu-nice.fr (A.C.); gaudart.a@chu-nice.fr (A.G.); cheddadi.y@chu-nice.fr (Y.C.); ruimy.r@chu-nice.fr (R.R.); 4CHU de Nice, Université Côte d’Azur, 06000 Nice, France; 5Inserm, C3M, Université Côte d’Azur, 06204 Nice, France

**Keywords:** slow-growing bacteria, urinary culture, urinary tract infections

## Abstract

*Aerococcus urinae* is an aerobic Gram-positive coccus that grows as tiny alpha-hemolytic colonies. *Actinotignum schaalii* is a slow-growing facultative anaerobic Gram-positive rod. These bacteria are part of the urogenital microbiota of healthy patients, but can also be involved in urinary tract infections (UTIs), particularly in elderly men and young children. Because *A. urinae* and *A. schaalii* are fastidious and are difficult to identify with phenotypic methods, they are underestimated causes of UTIs. Their growth is slow and requires a blood-enriched medium incubated under an anaerobic or 5% CO_2_ atmosphere for 48 h and from 24 to 48 h for *A. schaalii* and *A. urinae*, respectively. Furthermore, accurate identification is only possible using matrix-assisted laser desorption/ionization time-of-flight mass spectrometry (MALDI-TOF MS) or molecular-based methods. In rare cases, these bacteria can be responsible for invasive infections. We describe, here, an unusual case of bacteremic UTI caused by both *A. schaalii* and *A. urinae* in an 89-year-old woman. She presented with dyspnea, and bacteriuria was noted. This challenging clinical and microbiological diagnosis was made in our laboratory by Gram staining urine with a leucocyte count >50/μL and/or a bacterial count >14/μL urinary culture on a blood agar plate. After 10 days of antimicrobial treatment consisting of 2 g amoxicillin PO t.i.d., the patient was discharged with a complete clinical and biological recovery. *A. schaalii* and *A. urinae* are probably still underestimated causes of UTIs. Microbiologists could consider the presence of these two bacteria using appropriate culture and identification methods in cases where a positive direct examination of urine reveals small Gram-positive rods or cocci, where undocumented UTIs are present in elderly patients, but also where a urinary dipstick is negative for nitrites and is associated with leukocyturia.

## 1. Introduction

*Aerococcus urinae* and *Actinotignum schaalii* are part of the urinary microbiota [1,2,3] and have also been recently recognized as uropathogens in patients with certain underlying medical conditions [1,2,4,5]. The wider use of MALDI-TOF-MS technology means it is now possible to correctly identify these bacteria at the species level, which were formerly misidentified by biochemical methods [1,2]. These uropathogens have probably been underestimated as disease-causing agents due to the use of outdated identification methods, but also because bacteriological laboratories do not always use appropriate culture methods to isolate these slow-growing bacteria [1,2]. To our knowledge, pyelonephritis caused by both *A. schaalii* and *A. urinae* remains rare, especially in women [5,6,7]. We report here an unusual and interesting case in an 89-year-old woman who was successfully treated with amoxicillin acting on both bacteria.

## 2. Case Report

An 89-year-old woman was taken to the emergency ward of our university hospital from her nursing home with persistent drowsiness and dyspnea. Her medical history included non-insulin-dependent diabetes, hypothyroidism, and cognitive impairment. Upon physical examination, the patient was drowsy, but arousable, and had a Glasgow Coma Scale score of 15. Her body temperature was 37.9 °C. The patient was tachypneic, with a respiratory rate of 23 breaths/min, the oxygen saturation value was 86% on oxygen 3 L/min, and she had a quickSOFA score of one. Pulmonary auscultation revealed crackles in the left lower lung. Inflammatory markers showed elevated C-reactive protein (139 mg/L), but no leukocytosis (leukocyte count: 9.6 × 10^9^/L). There was no tenderness upon palpation of the kidney and the urinary bladder. UTI symptoms, such as dysuria, frequent urination, or urgent urination, were difficult to obtain because the patient suffered from cognitive impairment and remained silent throughout the examination. Urine dipstick showed traces of leukocytes and blood 3+, but no nitrites. A midstream urine sample was drawn for microbiological analysis, but no blood cultures were taken. SARS-CoV-2 and influenza tests were negative. Brain computed tomography showed no signs of cerebral hemorrhage. A chest X-ray showed bilateral perihilar infiltrates, and therefore, lower respiratory tract infection (LRTI) was first suspected, so she was given empiric antibiotherapy with 1 g of intravenous amoxicillin/clavulanic acid in the emergency ward. She was promptly transferred to the geriatric department, where two sets of blood cultures were finally drawn after the first dose of amoxicillin/clavulanic acid was given. The next day, urinalysis revealed 210 leukocytes/µL (iQ 2000, Beckman Coulter, Villepinte, France). In light of these first urinary results and after a review of the chest X-ray by a senior radiologist, the LRTI diagnosis was reconsidered, and she was switched from amoxicillin/clavulanic acid to intravenous cefotaxime to treat a presumed urinary tract infection (UTI). As the urinary leucocyte count was >50/μL, urine Gram staining was performed and showed Gram-positive cocci arranged in clusters. Urinary cultures remained sterile after 24 h of incubation on chromogenic agar plates (Uriselect 4^®^; Bio-Rad, Marnes-la-Coquette, France), but grew 10^6^ CFU/mL tiny alpha-hemolytic colonies after 48 h incubation on Columbia sheep blood agar under 5% CO_2_ (COL-S; Becton Dickinson, Le Pont-de-Claix, France). The MALDI-TOF MS of the colonies using a MicroFlexLT device and the BIOTYPER database (Bruker Daltonics, Wissembourg, France) successfully identified *A. urinae* (log score > 2). The antibiotic treatment was therefore changed to 1 g oral amoxicillin t.i.d. On the same day, the two anaerobic blood culture bottles (BCBs) incubated in a BacT/ALERT^®^ 3D system (BioMérieux, Marcy l’Etoile, France) were flagged as positive after incubation for 52 and 63 h, respectively. The aerobic BCBs were negative. Gram staining of the positive BCBs revealed Gram-positive cocci arranged in clusters and slightly curved Gram-positive rods (Figure 1).

MALDI-TOF identification from the positive BCBs, as previously described [8], failed on the first positive BCB, but matched with *A. urinae* with a maximum log score of 1.4 (the same identification, i.e., *A. urinae* was obtained for the first four scores) on the second one. The amoxicillin treatment was adjusted to 2 g t.i.d. for ten days for acute pyelonephritis associated with bacteremia. The two positive BCBs were subcultured on blood agar plates, and both grew tiny colonies after 48 h of anaerobic incubation. The two isolates were successfully identified at the species level using MALDI-TOF MS as *A. urinae* and *A. schaalii,* with maximum log scores of 2.07 and 2.12, respectively. We used E-test strips to determine antibiotic susceptibility. AST was interpreted in accordance with CASFM/EUCAST 2021 recommendations. The results are shown in Table 1. The patient was discharged after 8 days with complete clinical and biological recovery.

## 3. Discussion

*A. urinae* and *A. schaalii* are part of the urinary microbiota [1,2,3] and have also been recently recognized as uropathogens in patients with certain underlying medical conditions [1,2,4,5]. In 2005, Sturm et al. published the first case of nosocomial UTI caused by both *A. urinae* and *A. schaalii* in a male patient with an indwelling bladder catheter. The patient was finally cured after 7 weeks of the antibiotic treatment [5]. To our knowledge, pyelonephritis caused by both *A. schaalii* and *A. urinae* remains rare, especially in women [5,6,7]. We report, here, an unusual and interesting case in an 89-year-old woman who was successfully treated with amoxicillin acting on both bacteria.

The wider use of MALDI-TOF-MS technology means it is now possible to correctly identify these bacteria at the species level, which were formerly misidentified using biochemical methods: *A. schaalii* as *Gardnerella vaginalis, Arcanobacterium* spp., *Actinomyces meyeri* or *Actinomyces israelii*, and *A. urinae* as *Aerococcus viridans* or *Granulicatella* spp. [1,2]. Indeed, the recent studies showed that MALDI-TOF MS allows one to correctly identify *A. urinae* and *A. schaalii* strains at the species level with a high specificity and sensitivity [2,9,10]. These uropathogens have probably been underestimated as causes of disease due to the use of biochemical identification methods, but also because a few bacteriological laboratories employ the enriched medium required to grow these fastidious bacteria [2]. In our laboratory, urine culture protocols for slow-growing bacteria include a culture on Columbia sheep blood agar with incubation at 37 °C under 5% CO_2_ (COL-S; Becton Dickinson, Le Pont-de-Claix, France), and an anaerobic culture on Columbia CAP (Colistin + Aztreonam) blood agar (Oxoid) for 48 h, in addition to the usual chromogenic agar plates (Uriselect 4^®^; Bio-Rad, Marnes-la-Coquette, France), when the Gram stain is positive and yields Gram-positive rods or cocci. Gram staining is performed when the urinary leukocyte count is >50/μL and/or bacterial count is >14/μL. Our team has previously evaluated the potential interest of this protocol on 79,789 urinary samples for *A. schaalii*-related infections in a 3-year prospective study performed in our 1602-bed hospital. Finally, 35/79,789 (0.04%) urine samples yielded *A. schaalii* in a pure culture. Fourteen of the thirty-five (50%) patients with positive urine samples had *A. schaalii*-related UTIs [10]. This procedure could also be useful to identify new potential emerging uropathogens, such as *Alloscardovia omnicolens* or *Lactobacillus delbrueckii,* and facilitate the culture and identification of *Aerococcus* species. In the present case, urinalysis revealed 210 leukocytes/µL and the Gram stain revealed Gram-positive cocci arranged in clusters. The leukocyturia, clinical presentation, and chest final X-rays results eventually excluded the diagnosis of LRTI and suggested pyelonephritis.

The antibiotherapy was therefore switched to intravenous cefotaxime. *A. urinae* was then identified by a urinary culture, and the antibiotic treatment was changed to 1 g oral amoxicillin t.i.d., as *Aerococcus* species have very low MICs to β-lactams, making this class of antibiotics the treatment of choice against these pathogens [1]. In contrast, *A. urinae* is not consistently susceptible to the antibiotics frequently used to treat UTIs, such as fluoroquinolones (50–80%) [1,11], and is inherently resistant to sulfamethoxazole, which makes the action of the trimethoprim/sulfamethoxazole association uncertain [1]. *A. schaalii* is also consistently susceptible to aminopenicillin and is more frequently resistant to trimethoprim/sulfamethoxazole (60%) and second-generation quinolones (norfloxacin and ciprofloxacin) (99%) [2]. We should point out that we were unable to isolate *A. schaalii* in urine, and one explanation for this could be that *A. urinae* had grown over it. Another portal of entry seems unlikely considering that *A. schaalii* does not seem to be part of the gut microbiota [2], and the patient did not show any symptoms of a digestive disorder. Although *A. schaalii* can sometimes be involved in cellulitis and abscesses [2], cutaneous inoculation is also unlikely because the patient did not present with any skin wounds. Even if *A. schaalii* was not isolated in the urine sample, the patient had several predisposing factors for a UTI. Indeed, the advanced aged of the patient (89 years), the humid environment created by diapers, and urinary incontinence are all common risk factors for UTIs related to *A. schaalii* [2].

Interestingly, on the same day, *A. urinae* was isolated in a urine sample, and two anaerobic BCBs flagged as positive. The Gram staining of both BCBs showed slightly curved Gram-positive rods and Gram-positive cocci arranged in clusters. As *Corynebacteria* and *Propionibacterium* spp. are Gram-positive bacilli, and coagulase-negative *staphylococci* are Gram-positive cocci arranged in clusters, and both are frequently involved in BC contamination, our results may be attributed to BC contamination if the previous urine Gram stain had not showed Gram-positive cocci arranged in clusters and we had not simultaneously identified *A. urinae* in the urine. In our laboratory, we also performed the direct identification of BCBs, as our team described in a previously study [8], in which we compared the direct MALDI-TOF identification of BCBs (Day 0) with the identification of colonies on Day 1 (log (score) ≥ 2). We showed that using a log (score) ≥ 1.5 (with the same identification for the top three scores) on Day 0, we were able to correctly identify 100% of the staphylococci, enterococci, beta-hemolytic streptococci, *Enterobacterales,* and *Pseudomonas aeruginosa*. We did not test this identification protocol with any fastidious microorganisms, such as *A. urinae* and *A. schaalii*, because these bacteria are rarely involved in blood stream infections (BSI) [8]. Nevertheless, in the present case, the direct MALDI-TOF of one of the two anaerobic BCBs matched with *A. urinae,* with a log score of 1.4 (this identification was obtained for the first four scores). Although the maximum log (score) did not reach the threshold (log (score) ≥ 1.5), the combination of these results (Gram stain of BCB and direct MALDI-TOF of BCB and urinary culture) reassured us that the diagnosis of bacteremia caused by *A. urinae* was correct. Therefore, we immediately recommended optimizing the antibiotic dose to treat a BSI, and amoxicillin was adjusted to 2 g t.i.d. The tentative diagnosis of a BSI was confirmed 48 h later as both *A. urinae* and *A. schaalii* grew on the BCB subcultures. The patient was discharged after 8 days having completely recovered.

## 4. Conclusions

Pyelonephritis caused by both *A. urinae* and *A. schaalii* is rare especially in woman [5,6,7]. It is therefore important that this case is reported. The case is also interesting because its diagnosis presents a challenge to routine microbiology and to clinical practice, particularly for trainees. Microbiologists could consider the presence of *A. schaalii* or *A. urinae* in cases where a positive direct examination reveals small Gram-positive rods or cocci, where undocumented UTIs are present in elderly patients with an underlying disease or urinary incontinence, and also where a urinary dipstick is negative for nitrites and is associated with leukocyturia. The identification of these uropathogens is also important because they are inconstantly susceptible to trimethoprim/sulfamethoxazole and second-generation quinolones, which are widely used in the treatment of UTIs. An antimicrobial treatment with *β*-lactams is an efficient treatment and should be recommended. Finally, we would like to point out that our laboratory is operational 24 h/7, and the bacterial identification perfomed directly on BCBs allowed us to promptly diagnose an acute invasive infection and the physician to immediately optimize the antibiotic therapy.

## Figures and Tables

**Figure 1 microorganisms-11-02908-f001:**
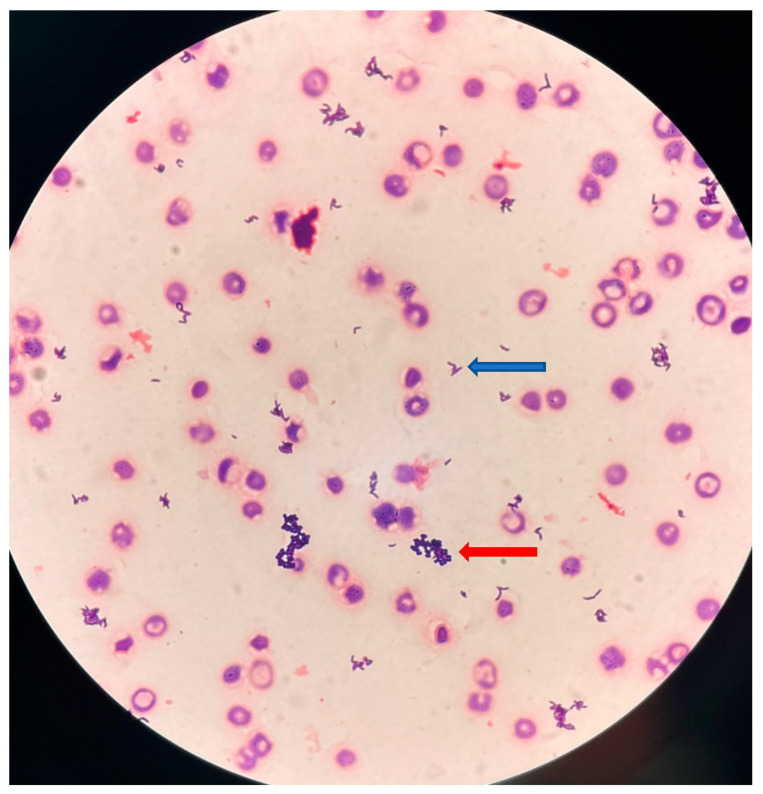
Gram staining (original magnification, ×500) of blood culture showing small, slightly curved, Gram-positive rods (blue arrow), and Gram-positive cocci arranged in clusters (red arrow).

**Table 1 microorganisms-11-02908-t001:** Results of antimicrobial susceptibility testing for *Actinotignum schaalii* and *Aerococcus urinae* isolates ^a^.

	*Actinotignum schaalii*		*Aerococcus urinae* ^c^	
Antimicrobial Agent	MIC (mg/L)	Susceptibility Categories ^b^	MIC (mg/L)	Susceptibility Categories
Amoxicillin	0.25	S	0.032	S
Amoxicillin-clavulanic acid	0.064	S		NA
Piperacillin-tazobactam	1	S		NA
Ciprofloxacin		NA	0.25	S
Levofloxacin		NA	1	S
Moxifloxacin	2	I		NA
Metronidazole	256	R		NA
Rifampicin		NA	0.064	S

^a^ Antimicrobial susceptibility testing (AST) were performed using *E*-test strips and usingCASFM/EUCAST 2021 breakpoints for anaerobic bacteria and *Aerococcus* spp., for *Actinotignum schaalii,* and *Aerococcus urinae*, respectively. ^b^ S, susceptible; R, resistant; I, susceptible, increased exposure; NA, not available. ^c^ The two strains of *Aerococcus urinae* isolated from urine and blood culture displayed the same AST results.

## Data Availability

Data are available upon reasonable request.

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
