# Peer review of "Acute Pyelonephritis with Bacteremia in an 89-Year-Old Woman Caused by Two Slow-Growing Bacteria: Aerococcus urinae and Actinotignum schaalii"

_microorganisms, 2023, doi:10.3390/microorganisms11122908_

Round 1
Reviewer 1 Report
Comments and Suggestions for Authors
This report describes a case of bacteremia caused by Aerococcus urinae and Actinotignum schaalii. These bacteria can be difficult to identify and to species determine. The report is of some interest and is relatively well-written. I have some concerns and some comments that I detail below:
Concerns
1. Abstract “Microbiologists should assess the presence of these two bacteria in urine using the appropriate culture and identification methods where direct examination is positive for small Gram-positive cocci or rods, and a urinary dipstick is negative for nitrites and associated with leukocyturia.”. I do not think that this case indicates that this statement is correct. This hypothesis would need testing in a prospective study. I believe that many additional bacteria would have been isolated but the clinical usefulness of the approach would still need to be investigated.
2. Introduction and conclusion “To our knowledge, pyelonephritis caused by both A. schaalii and A. urinae has been previously described only once, in a 78-year-old man presenting urinary incontinence [5].” I can expand your knowledge on this matter by pointing out that two such cases were described by Senneby (https://pubmed.ncbi.nlm.nih.gov/26838685/) and another six by Pedersen (https://pubmed.ncbi.nlm.nih.gov/27957598/).
3. Please explain if chest CT was performed and what that showed! Information on this is lacking and it seems to have been important to rule out pneumonia.
4. Table 1, please give MIC according to standard as 0.25 (not 0.19), 2 (not 1.5) and 1 (not 0.75). Use I for increased exposure according to EUCAST standard.
5. Discussion “A. urinae was then identified by urinary culture confirming the diagnosis of pyelonephritis”. This is not correct. The diagnosis of pyelonephritis is made by a combination of symptoms and signs with microbiology. Please change.
6. Discussion “Our results highlight the importance of (i) Gram staining a urinary sample where urinary dipstick is negative for nitrites and is associated with leukocyturia; and (ii) using an enriched culture medium when the Gram stain shows Gram-positive rods or Gram-positive cocci. This is the major take-home message of this case.” I realize that this is the way that your laboratory works but I think that the case does not highlight that this is important. The patient would have received a good antibiotic treatment for her condition without you working so much with the urine. The isolation of A. urinae and A. schaalii from blood was entirely sufficient for guiding treatment. This take-home message is not supported by the case.
7. Conclusions “Microbiologists should assess the presence of A. schaalii or A. urinae in cases where a positive direct examination”. “Should” is far too strong. Maybe “could consider”. I am of the opinion that the burden of proof is on the part that is suggesting a more complicated procedure. Such a procedure needs to be evaluated against the standard procedure before any conclusions about the usefulness of the procedure can be made.
8. Conclusion “UTIs. Antimicrobial treatment with aminopenicillin is the most efficient treatment and should be recommended.”. This claim is problematic in at least two ways. Firstly, it is not a conclusion of the present study and cannot be presented here. Secondly, there are no studies demonstrating that this particular treatment is more efficient than say cefotaxime. Therefore the claim must be deleted.
9. Conclusions “therapy, which was crucial for the patient’s recovery.”. This is entirely wrong. You cannot demonstrate any causality between the specific treatment and the outcome from one case. Delete!
Comments
1. Abstract “48 hours under anaerobic or 5% CO2 atmosphere.”. This is not entirely correct. A. urinae can often be isolated after 24 hours of growth.
2. Abstract “presented dyspnea and asymptomatic bacteriuria.” is not correct. She presented with dyspnea and bacteruria was noted. Please change.
3. Abstract “direct bacterial identification on positive blood culture bottles (BCBs) by MALDI-TOF MS.” I do not think that this contributed to the diagnosis in a meaningful manner. Delete.
4. Introduction and discussion “means it is now possible to correctly identify to the species level these bacteria,” should be “means it is now possible to correctly identify these bacteria to the species level,”.
5. Please mention if the patient was tender upon palpation of the kidney or the urinary bladder or if she experienced symptoms from the urinary tract.
6. Case presentation, what is “Brain tomodensitometry”?
7. Case presentation, “two sets of blood cultures were finally drawn.” Please specify if this was after one dose of antibiotics.
8. Case presentation, “(and four times repeatable)”, what does this mean? I do not understand..
9. Case presentation, how was species determination performed on the isolates from the positive blood cultures? Method and results should be given.
Comments on the Quality of English Language
Some improvements can be made
Reviewer 2 Report
Comments and Suggestions for Authors In the case study "Acute Pyelonephritis with Bacteremia in an 89-Year-Old Woman Caused by Two Slow-Growing Bacteria: Aerococcus urinae and Actinotignum schaalii" the authors decribe a rare case.The case is interesting. Yet the authors need to address the following:
- More background about the previous case (the first reported one) should be provided.
- The authors need to discuss more the sensitivity and specificity of the selected method of identfication related to these two species.
Minor:
The names of the bacterial species are not written in italics in more than one instance. Please fix al
Comments on the Quality of English Language
Minor revisions are required
Reviewer 3 Report
Comments and Suggestions for Authors
In this case report, it was well described that Aerococcus urinae and Actinotignum schaalii were the causes of acute pyelonephritis.
However, there were already many studies about Aerococcus urinae and Actinotignum schaalii for UTIs.
Are there any updates compared to other studies or case reports in the Aerococcus urinae and Actinotignum schaalii papers?
